# *Trypanosoma cruzi* Presenilin-Like Transmembrane Aspartyl Protease: Characterization and Cellular Localization

**DOI:** 10.3390/biom10111564

**Published:** 2020-11-17

**Authors:** Guilherme C. Lechuga, Paloma Napoleão-Pêgo, Carolina C. G. Bottino, Rosa T. Pinho, David W. Provance-Jr, Salvatore G. De-Simone

**Affiliations:** 1Center for Technological Development in Health/National Institute of Science and Technology for Innovation on Diseases of Neglected Population (INCT-IDPN), FIOCRUZ, Rio de Janeiro 21040-900, Brazil; guilherme.lechuga@yahoo.com.br (G.C.L.); paloma.pego@cdts.fiocruz.br (P.N.-P.); carolina.bottino.bio@gmail.com (C.C.G.B.); bill.provance@cdts.fiocruz.br (D.W.P.-J.); 2Cellular Ultrastructure Laboratory, FIOCRUZ, Oswaldo Cruz Institute, Rio de Janeiro 21040-900, Brazil; 3Interdisciplinary Medical Research Laboratory, FIOCRUZ, Oswaldo Cruz Institute, Rio de Janeiro 21040-900, Brazil; 4Clinical Immunology Laboratory, FIOCRUZ, Oswaldo Cruz Institute, Rio de Janeiro 21040-900, Brazil; rospinho@ioc.fiocruz.br; 5Department of Molecular and Cellular Biology, Federal Fluminense University, Niterói 24220-008, Brazil

**Keywords:** *Trypanosoma cruzi*, presenilin, aspartic protease, SPOT-synthesis, anti-peptide antibodies, immunolocalization, transmembrane domains

## Abstract

The increasing detection of infections of *Trypanosoma cruzi*, the etiological agent of Chagas disease, in non-endemic regions beyond Latin America has risen to be a major public health issue. With an impact in the millions of people, current treatments rely on antiquated drugs that produce severe side effects and are considered nearly ineffective for the chronic phase. The minimal progress in the development of new drugs highlights the need for advances in basic research on crucial biochemical pathways in *T. cruzi* to identify new targets. Here, we report on the *T. cruzi* presenilin-like transmembrane aspartyl enzyme, a protease of the aspartic class in a unique phylogenetic subgroup with *T. vivax* separate from protozoans. Computational analyses suggest it contains nine transmembrane domains and an active site with the characteristic PALP motif of the A22 family. Multiple linear B-cell epitopes were identified by SPOT-synthesis analysis with Chagasic patient sera. Two were chosen to generate rabbit antisera, whose signal was primarily localized to the flagellar pocket, intracellular vesicles, and endoplasmic reticulum in parasites by whole-cell immunofluorescence. The results suggest that the parasitic presenilin-like enzyme could have a role in the secretory pathway and serve as a target for the generation of new therapeutics specific to the *T. cruzi*.

## 1. Introduction

Chagas disease is caused by the flagellate protozoan *Trypanosoma cruzi* and endemic to Latin America, currently affecting 8 million people worldwide [1]. Despite successful governmental strategies to control *Triatoma infestans*, the primary vector responsible for transmission in Southern Cone countries [2], the emergence of secondary vector species, and multiple instances of oral outbreaks, underlie recent increases in transmission [3,4]. Alarmingly, current migratory trends out of endemic areas have greatly contributed to the spread of the disease to non-endemic countries that are causing social issues and having a high economic impact on national health care systems, most notably in North America and Europe [5].

There are only two approved drugs available for the treatment of Chagas disease, nifurtimox and benznidazole (Bz). They are only indicated for the acute phase and are ineffective during the chronic phase of the disease. Additionally, they produce severe side effects that often lead to a discontinuation of treatment before the prescribed endpoint [6,7]. Other concerns are naturally resistant strains [8] and a dormancy state of *T. cruzi* that also leads to resistance [9]. Recent clinical trials on the effectiveness of Bz for preventing or slowing cardiac pathogenesis showed unsatisfactory results [10]. Similar results were observed with posaconazole [11,12], which reinforces the need to search for new, more efficient therapeutic drugs.

A promising approach to identify new targets is to investigate parasite metabolism pathways. Among the many metabolic candidates explored in different *T. cruzi* life stages, proteases have been intensively studied for inhibition [13,14,15]. These proteins are involved in basic functions like nutrition and cell division, as well as other more specific activities such as evading the immune system or acting as virulence factors [16,17,18]. Cruzipain, the major cysteine protease of *T. cruzi*, has already been the aim of multiple studies for the development of new inhibitors [19,20]. Numerous other proteases could also be targets. According to the MEROPS protease database on those expressed from the *T. cruzi* genome, cysteine, and metalloproteases represent the most abundant classes with more than 150 different annotations, whereas serine, threonine, and aspartyl proteases are present in a lesser number [21].

The presence and role of aspartic proteases in *T. cruzi* is an understudied area of its biology. Our group previously identified the presence of two distinct aspartyl proteases activities that differed by their cellular localization. One activity was detected in the supernatant of whole parasite extracts after centrifugation at 100,000× *g*, while the other was associated with a membrane-enriched fraction [22]. The presence of an aspartic enzyme associated with the membrane drew our attention for its potential to have a presenilin (PS)-like function as the catalytic portion of a parasitic γ-secretase complex (GSC). This complex is a multimeric intramembrane structure found in many species ranging from animal to plants. It has been localized in different subcellular membrane compartments (e.g., mitochondria, cell membrane, and nuclear envelope) with a major distribution to the endoplasmic reticulum (ER) and Golgi apparatus [23,24,25,26]. While PS-like enzymes have been confirmed in the parasites *Schistosoma mansoni* [27] and *Plasmodium falciparum* [28,29] as well as the nematode *Caenorhabditis elegans* [30], its presence in *T. cruzi* has not been previously described. The importance of the PS-like aspartyl protease in *P. falciparum* for its invasion of red blood cells [28] suggested that a *T. cruzi* form could also serve a vital function.

Here, genomic information was used to generate a peptide library of the *T. cruzi* PS-like coding region to represent potential linear B-cell epitopes. Two of the multiple linear B-cell epitopes identified with Chagas patient sera were used to generate rabbit monospecific antibodies for the cellular localization of the enzyme by fluorescence microscopy. Bioinformatics and biochemical approaches were used to characterize the enzyme structure. The identification of this novel transmembrane aspartyl-protease, located mainly in the flagellar pocket, opens the potential to elucidate its metabolic function and role in the *T. cruzi* homeostasis.

## 2. Materials and Methods

### 2.1. Reagents

Amino-PEG500-UC540 cellulose membranes were obtained from Intavis AG Bioanalytical Instruments (Cologne, Germany). Amino acids for peptide synthesis were purchased from Calbiochem-Merck (Darmstadt, Germany). BSA, acetic anhydride, N, N-dimethylformamide, Freund’s incomplete adjuvant, DAPI, TRITC, and FITC labeled anti-rabbit IgG antibodies, TRITC-phalloidin, monodancylcadaverine, maleimide activated kit, Tween^®^ 20, acetonitrile, monodancylcadaverine, tissue protease inhibitor cocktail, and trifluoracetic acid and were obtained from Sigma-Merck (St, Louis, MO, USA). Rabbit and goat alkaline phosphatase-labeled anti-human-IgG (AP-anti-huIgG) and anti-rabbit IgG (AP-anti-rabIgG) were purchased from Abcam (Cambridge, MA, USA). Super Signal R West Pico chemiluminescent substrate was from Pierce Biotechnology (Rockford, IL, USA). Centrifugal Filter Units (cutcoff 10 kDa) were from Millipore (Bedford, MA, USA) and Nitro-Block II from Applied Biosystems (Foster City, CA, USA). Fetal bovine serum (FBS) was from Thermo Fisher Scientific Inc (Waltham, MA, USA). Brain and heart infusion (BHI) medium from Difco and nitrocellulose membrane from BioRad (Hercules, CA, USA).

### 2.2. Parasites and Cell Culture

The CL-Brener strain of *T. cruzi* was obtained from the Trypanosomatidae collection (CT–IOC/05) curated by Dr. Maria A. de Souza (Oswaldo Cruz Institute-FIOCRUZ). Epimastigote parasites were propagated in BHI medium supplemented with 10% FBS [31]. Trypomastigote forms were obtained from T. cruzi-infected Vero cell cultures, maintained in RPMI-1640 medium supplemented with 10% FBS, on the 4th day post-infection. Intracellular amastigotes were obtained by trypsinization of infected Vero cells monolayers and disruption of cells using a 25-gauge needle followed by centrifugation (3000× *g* for 15 min at 25 °C) [32].

### 2.3. Parasite Extract

*T. cruzi* epimastigotes in the log phase (4th day) were washed 3 times in PBS (pH 7.2) by centrifugation (5000× *g* for 30 min at 4 °C). The final parasite pellet was suspended in 150 μL of extraction buffer (150 mM NaCl, 50 mM Tris 50 (pH 7.5) with 1% Triton X-100 and protease inhibitors) and subjected to 6 cycles of freeze-thawing. After centrifugation (10,000× *g* for 1 h at 4 °C), the detergent soluble fraction was collected. Protein concentration was measured using the Folin-Lowry method.

### 2.4. Synthesis of the SPOT Peptide Array on Cellulose Membrane

The DNA sequence for the putative PS-like aspartic peptidase (Q4CMV5) of the CL Brener strain of *T. cruzi* was retrieved from the National Center for Biotechnology Information database. A library of 14 amino acid peptides with a 9-amino acid overlap was designed to represent the entire coding region (372 aa) of the PS-like protein and automatically synthesized onto cellulose membranes using an Auto-Spot Robot ASP222 (Intavis, Koeln, Germany) according to the SPOT synthesis protocol [33,34]. Coupling reactions were followed by acetylation with acetic anhydride (4%, *v*/*v*) in *N*,*N*-dimethylformamide to render peptides unreactive during the subsequent steps. After acetylation, F-moc protective groups were removed by the addition of piperidine to render the nascent peptides reactive. The consecutive amino acids were added by this same process of coupling, blocking, and deprotection until the desired peptide was generated. After the addition of the last amino acid in the peptide, amino acid side chains were deprotected using a solution of dichloromethane–trifluoracetic acid–tri isobutyl silane (1:1:0.05, *v*/*v*/*v*) and washed with methanol. Membranes containing the synthetic peptides were either probed immediately or stored at −20 °C until needed.

### 2.5. Screening of SPOT Membranes

SPOT membranes were washed with TBS (50 mM Tris-buffer saline, pH 7.0) and then blocked with TBS-MT (Tris-buffer saline, 3% defatted milk, 0.1% Tween^®^ 20, pH 7.0) under agitation for 1 h at room temperature or overnight at 4 °C. After extensive washing with TBS-T (Tris-buffer saline, 0.1% Tween 20, pH 7.0), membranes were incubated for 2 h with human patient sera (1:250 dilution in TBS-MT), washed 3 times with TBS-T, incubated for 1 h with AP-anti-huIgG (1:5000 dilution in TBS-MT). Next, membranes were washed 3 times with TBS-T, and then the buffer exchanged to CBS (50 mM citrate-buffer saline) before the addition of the chemiluminescent enhancer Nitro-Block II. The chemiluminescent substrate Super Signal R West Pico was applied, and signals were immediately detected by an MF-ChemiBis 3.2 (DNR Bio-Imaging Systems, Neve Yamin, Israel) as described previously [35]. Briefly, a digital image file was generated at a resolution of 5 MP and the signal intensities quantified using TotalLab (Nonlinear Dynamics, Durham, NC, USA) software.

### 2.6. Peptide Synthesis and BSA and Biotin Conjugation

Two *T. cruzi* peptides, EP8 and EP9 (Table 1), were chosen to be synthesized by the F-moc strategy in a synthesizer machine (PSS-8, Schimadzu, Kyoto, Japan) with a C-terminal cysteine that was used to conjugate peptides to bovine serum albumin (BSA) using a maleimide activated kit according to manufacturer’s instructions. The reaction mixture was passed through a centricon-P10 and the peptide concentration in the filtrate (uncoupled peptide) measured on a Qubit device (Thermo Fisher, Waltham, MA, USA). Efficiency was calculated as (total peptide-uncoupled peptide/total peptide), which ranged between 80% and 85% in every case. Peptides were also conjugated to biotin in the C-terminal region, as previously described [36].

### 2.7. Rabbit Polyclonal Antibodies Production

Two New Zealand rabbits were immunized by subcutaneous injection of peptide-BSA (150 µg) emulsified with an equal volume of Freund’s incomplete adjuvant. Three other inoculations, without adjuvant, were administered each 7 days later and the serum was collected 5 days after the last injection. Blood was collected under standard bioethics conditions from the marginal ear vein.

To remove BSA-reactive antibodies, the rabbit sera anti-EP8 (rab-EP8) was passed over a Sepharose-4B column (3 cm × 1 cm i.d.) that was coupled with BSA according to conditions previously described [37]. The removal of anti-BSA antibodies was evaluated by the loss of reactivity in the rab-EP8 to BSA by immunoblot (Appendix A).

### 2.8. Parasite Sample Preparation, SDS-PAGE, and Immunoblotting

The expression of *T. cruzi* presenilin-like was evaluated in log phase epimastigotes (5 × 10^6^/mL) that were washed 3 times in PBS (pH 7.2) and incubated for 24 h in BHI in the presence or absence of FBS (10%). In addition, parasites (5 × 10^6^/mL) were incubated with 100 and 200 µM of gamma-secretase inhibitors, DAPT, and Compound XXI/E, respectively, for 24 h. After incubation, parasites were washed 3 times in PBS, and a soluble fraction was prepared as described previously.

Protein samples were separated by sodium dodecyl sulfate (SDS)-polyacrylamide gels electrophoresis (SDS-PAGE, 10%) under reducing conditions. Proteins bands were visualized with Coomassie Brilliant Blue R-250 stain.

For immunoblot analysis, total protein (20 μg) was separated by SDS-PAGE and transferred to a nitrocellulose membrane (2 µm). After blocking with TBST buffer (TBS, 50 mM Tris-Cl, 150 mM NaCl and 0.05% tween 20, pH 7.5) containing 5% skim milk, membranes were incubated with primary serum (1:200) overnight at 4 °C, washed three times in TBS-T and incubated with the AP-anti-rabIgG (1:5000) for 1 h at 25 °C. After washing, the immunolabeled proteins were detected by chemiluminescence using SuperSignal West Pico substrate kit. The densitometry of the bands was performed using Image J (http://rsbweb.nih.gov).

### 2.9. Absorption of Proteins to Pepstatin A-Agarose

Epimastigote was harvested by centrifugation (3000× *g* for 15 min at 4 °C) in the log phase (4th day of cultivation) and washed 3 times in ice-cold PBS (pH 7.2) through centrifugation (3000× *g*, 15 min at 4 °C). Next, parasites were disrupted by 6 cycles of freeze-thawing (−80 °C/37 °C) in 0.1 M sodium acetate buffer (pH 3.5) followed by centrifugation (100,000× *g* for 60 min at 4 °C). The supernatant was removed, and the pellet was extracted in AC buffer (0.1 M acetate buffer, (pH 3.5) with 1% CHAPS) for 1 h on ice. After a 2nd ultracentrifugation (100,000× *g* for 1 h at 4°C), the detergent fraction in the supernatant was applied to a pepstatin-A agarose column (10 cm × 1 cm I.D.) previously equilibrated with AC buffer at a flow rate of 15 mL h^-1^. After washing with 10 bed volumes of AC buffer, bound proteins were eluted with 1 M NaCl, 0.1 M Tris-HCl (pH 8.6) buffer.

Alternativelly, *T. cruzi* epimastigote cell lysates (40 µg of protein) extracted with Triton X-100 (1%) were incubated in the presence of 40 µL of a pepstatin A-agarose slurry (Sigma) in STE buffer (150 mM NaCl, 50 mM Tris, 2 mM EDTA, pH 7.4), overnight at 4 °C. Samples were centrifuged at 700× *g* and unbound fraction was collected. Then, the remaining unbound proteins were removed, by washing 3 times with equilibration buffer, and beads incubated with Laemmli buffer followed by boiling at 100 °C for 5 min. Samples were analyzed by SDS-PAGE and Western blotting.

### 2.10. Enzyme-Linked Immunosorbent Assay (ELISA)

Wells of a 96 well-plate (C96 Microwell, Nunc, New York, NY, USA) were loaded with the designated synthetic peptide (1 µg/well) in coating buffer (Na_2_CO_3_–NaHCO_3_, pH 9.6) overnight at 4 °C. After washing (3×) with PBS-T (PBS, pH 7.2 with 0.1% Tween^®^ 20), wells were blocked with PBS-M (PBS, pH 7.2 with 2% defatted milk) for 2 h at 37 °C. Next, a dilution series of immunized rabbit sera (1:10, 1:100, 1:200, 1:400, 1:800 and 1:1000 in 50 µL of PBS, pH 7.2) was added for 2 h at 37 °C. Following several washes with PBS-T, the plates was incubated with alkaline phosphatase labeled goat anti-rabbit IgG (1:5000 in 50 µL of PBS, pH 7.2) for 2 h. Wells were washed with PBS-T before the addition of p-nitro phenylphosphate (pNPP) substrate. After 15 min, a stop solution (3N NaOH) was added, and the absorbance was measured at 405 nm within 2 h using a FlexStation 3 Microplate Reader (Molecular Devices, Sumyvale, CA, USA).

### 2.11. Immunofluorescence Microscopy

Parasites in different life stages (epimastigotes, trypomastigotes, and amastigotes) were obtained as described above and treated with paraformaldehyde (4%) in suspension for 5 min. Parasites were collected by centrifugation (3000× *g* for 5 min at RT) and washed 3× in PBS. Next, 5 × 10^6^ parasites were placed onto a glass slide and allowed to air-dry overnight. Next, cells were permeabilized with 0.1% Tween^®^ 20 in PBS (pH 7.4) for 5 min and blocked with 1% casein in PBS (pH 7.4) for 15 min at 37 °C. Afterwards, parasites were incubated with rabbit anti-EP8 or pre-immune (1:100), for 2 h at 37 °C. Slides were washed in PBS and then incubated with TRITC-labeled anti-rabIgG (1:400) for 1 h at 37 °C. Next, parasites were washed in PBS and stained with DAPI, and mounted in DABCO solution. An Axio Imager M2 fluorescence microscope (Carl Zeiss, Oberkochen, Germany) was used to collect a z-stack of images.

To label autophagic vacuoles, epimastigotes (5 × 10^6^/mL) were cultivated for 24 h in BHI medium deprived or supplemented with 10% FBS. After incubation, parasites were washed twice in PBS and incubated with 50 µM of monodancylcadaverine (MDC) for 30 min. Parasites were washed with PBS to remove the dye and visualized under fluorescence microscope.

To evaluate co-localization of EP8 with intracellular organelles, epimastigotes (5 × 10^6^) were cultivated as described above and washed twice with PBS, parasites were incubated for 15 min with 0.05 mg/mL of DiOC_6_(3) to probe the ER and 30 min with BODIPY FL–pepstatin A (1 µg/mL). To specifically probe the flagellar pocket, epimastigotes were incubated with 10 µg/mL Concanavalin A-FITC (ConA-FITC) for 30 min at 4 °C [38]. After incubation, parasites were washed 3 times with PBS and fixed with paraformaldehyde (4%), and processed for anti-EP8 labeling as described above. For image analysis, fluorescence was analyzed in Image J using BAR plugin for a multichannel plot profile.

### 2.12. Database Searches, Computational, and Phylogeny Studies

Searches for possible domains and structural components characteristics of PS in *T. cruzi* were performed on the Uniprot database (http://www.uniprot.org/) based on sequence homologies with similar proteins identified in other organisms. Sequence alignments were conducted on the Clustal Omega server (http://www.ebi.ac.uk/Tools/msa/clustalo/).

The potential transmembrane domains (TMD) of the *T. cruzi* enzyme were analyzed by 3 different predictions programs; the TMpred [39], the TopCons [40], and MemConP [41]. The model most consistent with the identified epitopes and motifs was obtained using TopCons, which used an algorithm based on the statistical analysis of base, and the prediction was made using a combination of several weight-matrices for scoring.

Secondary structure predictions were obtained from PSIPRED (http://bioinf.cs.ucl.ac.uk/psipred/) and CDM (http://gor.bb.iastate.edu/cdm/) servers. The tertiary structure prediction was performed on the LOMETS server (http://zhanglab.ccmb.med. umich.edu/LOMETS/) and I-TASSER (https://zhanglab.ccmb.med.umich.edu/I-TASSER/; data not shown). For protein–protein interactions, a network analysis was performed using STRING and Cytoscape (version 3.7.2) [42].

For curation of a family of target sequences for Q4CMV5, a set of aspartic proteases sequences were retrieved from the Uniprot database using the following criteria: An EC designation of 3.4.23 for aspartic endopeptidases; a length between 300 to 800, and a reviewed annotation. Another set of sequences homologous to other proteins, which have been cited by Blast prospected, was added after a delta-blast of the Refseq database with coverage greater than 80% of their sequence-issue and identity over 10–30%. From this set, an alignment was performed on the Mafft server with blossum80 and a gap penalty of 2.5, followed by clustering to eliminate the excessive number of gaps with the method “minimum linkage”.

Multiple sequence alignments and the cladogram were generated using Clustal Won sequences with following accession numbers: *Acromyrmex echinatior* (F4 × 309), *Angiostrongylus cantonensis* (C7BVX5), *Arabidopsis thaliana* (Q0WT05), *Ascaris suum* (F1L7 × 7), *Bathycoccus prasinos* (K8EKK4), *Bombyx mori* (I6TQW0), *Bos taurus* (Q9XT97), *Caenorhabditis elegans* (O02100), *Camponotus floridanus* (E2AIY0), *Canis familiaris* (Q6RH31), *Cricetulus griséus* (G3HNF2), *Danio rerio* (Q9W6T7), *Desmodus rotundus* (K9IKI1), *Entamoeba histolytica* (C4M1A2), *Gallus gallus* (Q4JIM4), *Heterocephalus glaber* (G5BP42), *Homo sapiens* (P49768), *Macaca fascicularis* (Q8HXW5), *Macaca mulatta* (F7AGU8), *Microcebus murinus* (P79802), *Mus musculus* (P49769), *Mus musculus* (Q3UYK2), *Mustela putorius furo* (G9KIX0), *Pan troglodytes* (K7CVQ1), *Pongo abelii* (Q5R780), *Rattus norvegicus* (P97887), *Sus scrofa* (Q0MS44), *Xenopus laevis* (O12976), *Trichomonas vaginalis* (A2DZ73), *Leishmania mexicana* (E9AQF0), *Leishmania major* (Q4QF26), *Leishmania infantum* (A4HWP2), *Leishmania braziliensis* (A4H8C1), *Trypanosoma cruzi* (Q4E0Z2), *Trypanosoma brucei brucei* (Q38F54), *Trypanosoma brucei gambiense* (C9ZXP4), *Trypanosoma congolense* (G0UTQ1), *Trypanosoma vivax* (G0U246).

### 2.13. Ethics Statement

Approval for the experimental use of animals was granted by the Ethics Committee for Experimentation on Animals of the Oswaldo Cruz Foundation (CEUA n^o^ P-0279/06), Rio de Janeiro, before the start of the study. Animals were housed and maintained according to the institutional guidelines for animal studies, which conformed to the specifications outlined in the US National Institutes of Health guidelines for the care and use of laboratory animals. All efforts were made to minimize suffering.

### 2.14. Statistical Analysis

Statistical analysis was performed using GraphPad Prism version 5.0. The statistical difference using a t-test was considered if *p*-value ≤ 0.05.

## 3. Results

### 3.1. Spot Synthesis and Epitope Identification

The identification of the antigenic regions that were present in the PS-like(Q4CMV5) of *T. cruzi* was performed through the Spot Synthesis technique, which can define linear B cell epitopes. An array of 14-mer peptides with a 9 amino acid overlap was designed that represented the entire coding region of Q4CMV5 and was synthesized in situ on a cellulose membrane. The membrane was incubated with a pool of sera from patients (n = 8) whose diagnosis for Chagas disease was confirmed. An image of the detected chemiluminescent signal is shown in Figure 1A. The signals were quantified and normalized to the highest intensity to determine the relative intensity percentage, which was plotted against their positions in Figure 1B. The individual peptide sequences and their locations of the membrane are listed in Figure 1C. From the 73 peptides in the library, 20 peptides showed signal intensities above 50% that was considered above the cutoff for a positive reaction (summarized in Table 1).

### 3.2. Structure and Topology of *T. cruzi* PS-Like

All eukaryotic PS family members are predicted to be multi-pass membrane proteins with several transmembrane domains (TMDs). Using the TopCons program, a topology with nine TMD was predicted for the *T. cruzi* PS-like protein from the translated genomic sequence (Figure 2). It also indicated that the N- and C-terminal should be present in the cytosol and luminal spaces, respectively. A large intracellular hydrophilic loop was identified between TMD 6 and TMD 7. The overall topology of *T. cruzi* PS showed a striking similarity to the structure of other previously described PSs proteins, with the same number of TM domains. The model presented in Figure 4 also shows the localization of the epitopes identified by the SPOT synthesis analysis concerning the TMDs. From the 11 epitopes identified, 7 (EP2, EP3, EP5, EP6, EP7, EP9, and EP10) were located in coil/loop structures of the PS protein with EP2, EP3, and EP11 appearing on the extracellular surface of the plasma membrane while the others epitopes were on the internal cytosolic face. Surprisingly the EP1, EP4, EP9, and EP11 were located bordering the membrane-extracellular interface region of the protein. In addition, predicted protein topology revealed possible intracellular domains for endoproteolytic cleavage between TMD 6 and 7 (Figure 2).

### 3.3. Motifs Crucial for A22 Aspartyl Protease Family and γ-secretase Activity Are Conserved in *T. cruzi* Homologs

The data available from studies on the γ-secretase complexes in mammalian cells have indicated that certain amino acid motifs are crucial for its proteolytic activity, substrate recognition, and complex assembly [43]. To prove that these critical motifs are also conserved in the putative *T cruzi* PS-like, multiple sequence alignments were performed utilizing amino acid sequences of PS component homologs from *A. thaliana*, *Physcomitrella patens*, *Chlamydomonas reinhardtii*, *Dictyostelium discoideum*, and *Homo sapiens* (data not shown). Potential homologs were identified by PSI-BLAST and the level of similarity among the PSs from these species was aligned to the *Homo sapiens* sequence (data not shown).

By alignment of the *T. cruzi* PS-like sequence (Q4CMV5) with the *H. sapiens* PS protein (data not shown), we found several compatible motifs sequences (Table 2). The human ATIKS motif was associated with the sequence WSVLN (aa 34–38) from the *T. cruzi* PS-like, both localized in TMD1. The human SILNAAIMIS motif was identified with the SIVNALILVA (aa 70–79) motif in the *T. cruzi* PS-like protein, both present in the TMD2. The AxxxS (AQRDS) present in the TMD2 of the *H. sapiens* PS1 protein was correlated to the *T. cruzi* PS-like protein NSSND sequence (aa 256–260) located in the intracellular loop between TMD6 and TMD7. Conversely, the SxxxA (SALMA) motif present in the TMD5 of both human PS1 and PS2 proteins aligned with SVIVG (aa 168–172) in *T. cruzi* PS-like also located in TMD5. The GxGD motif aligned with PFKLGLGD (aa 283–287) in the *T. cruzi* PS-like, in TMD7 domain. As in human PS1, highly conserved domains in the catalytic pocket (YD; 192–194 and GLGD; 287–290) were identified closely in TM6 and TM7, respectively.

The human TMD8 has two motifs [GxxxG (GVKLG) and SxxxGxxxxA (SVLVGKASA)], which lie on the side of the catalytic Asp. Both motifs contain sites of FAD mutations, which form a portion of the catalytic core of the PS and influence helix packing that may modulate enzymatic activity. In the *T. cruzi* PS-like, they are both situated in TMD7, observed as PFKLG (aa 283–287) and SVLSARAAL (aa 295–303). The TMD8 and TMD9 of the human protein have two unusual, sequential motifs: AxxxAxxxG (ACFVAILIG) and AxxxSxxxG (ALPALPISITFG). These motifs remained in similar positions, the *T. cruzi* PS-like, the sequence ASTVAVCFG was located in TMD8 (aa 312–320) and ALPALPISICFG (336–347) in the TMD9. Finally, the conserved C-terminal PALP motif that determines the conformation of the active site [44,45,46,47] was identified at TMD9 (aa 339–342) in *T. cruzi* sequence (Table 2 and Figure 2).

### 3.4. Phylogeny and Protein–Protein Interaction

A group of sequences for PSs was retrieved on the Uniprot server (http://www.uniprot.org) through a search using the criteria name, “PS”, and a length of 300 to 800. A phylogenetic tree was created with the sequences that was aligned by the neighbor-joining algorithm using a CLUSTAL W program (Figure 3A). The *T. cruzi* PS-like aspartyl proteases were localized in a separate subgroup along with the *T. vivax* enzyme that was detached from the other protozoa subclasses of this protease family, like *Leishmania* sp. and *T. brucei*. Network analysis of predicted protein–protein interaction revealed that *T. cruzi* PS-like could interact with calreticulin, signal peptide peptidase (SPP), peptidase, formin, protein kinase, and glycogen synthase kinase 3A (GSK3A) (Figure 3B).

### 3.5. Antisera Production, SDS-PAGE and Western Blotting

*T. cruzi* PS-like transmembrane protein is expected to be identified in membrane preparations for that we separated a soluble and detergent fraction using CHAPS (1%) and performed a pepstatin-A affinity chromatography. Western blot using anti-detergent fraction immunized rabbit sera and a pool of sera from chagasic patients revealed some reactive bands of approximately 70, 67, 45, 43, 36, and 23 kDa (Appendix A). To further investigate these findings, we performed the Spot-synthesis that identified 11 linear B-cell epitopes in the *T. cruzi* PS-like coding region, subsequently analyzed by several physical-chemical parameters that included a prediction of stability, net charge in a neutral buffer and hydrophobicity to evaluate their potential to serve as antigens for antibody production. In addition, the sequences were compared to the translated nucleotide databases of other parasites, in particular *Leishmania* sp., to choose peptides specific for *T. cruzi* with the lowest chance to display cross-reactivity. Lastly, the uniqueness of each sequence to only represent the *T. cruzi* PS-like protein was evaluated to ensure specificity. Two epitopes, designated EP8 and EP9, were chosen for the production of anti-sera in rabbits.

Both epitopes were synthesized as peptides with a cysteine at the C-terminal to permit their conjugation to BSA through maleimide chemistry. The peptide-BSA preparations of EP8 and EP9 were used to inoculate rabbits, and the resulting polyclonal serum was refined through a Sepharose-BSA column to remove anti-BSA antibodies. The loss of reactivity to BSA was accessed by Western blot with a membrane containing two concentrations of BSA (10 and 20 µg) that suggested that the BSA reactivity in anti-EP8 serum was eliminated (Appendix A). In preliminary immunofluorescence experiments with a dilution series of the two anti-peptide sera, anti-EP8 serum maintained a positive reaction when diluted up to 1:200 in immunofluorescence tests compared to 1:50 for EP9 (data not shown). Additionally, peptide ELISAs confirmed the higher performance of the anti-EP8 sera as well as a competition assay that showed a sharp inhibition activity by preincubation with an increasing concentration of EP8 peptide. An inhibition of 50% was obtained with 20 µM of peptide (Appendix A). Based on these results, the rabbit antisera anti-EP8 was chosen for further characterization of the *T. cruzi* PS-like protein.

Western blot analysis of epimastigote whole extract revealed 4 major bands, a stronger band at ~23 kDa, and others at ~32 kDa, 40 kDa, and 76 kDa (Figure 4A). To test if PS-like protein bind to its specific inhibitor pepstatin A, parasite lysate was adsorbed to pepstatin A-agarose and bound proteins analyzed by Western blot using anti-EP8 sera that identified bands of ~23 kDa and ~32 kDa (Appendix A).

To examine the pattern of *T. cruzi* PS-like expression in epimastigotes, parasites were treated with gamma-secretase inhibitors (GSI (DAPT and XXI/E) and under serum deprivation. Treatment of parasite with GSI decreased presenilin holoprotein (~41 kDa) expression (Figure 4B), while it increased presenilin C-terminal fraction (CTF) (Figure 4D). In addition, FBS deprivation increased significantly (approximately 9-fold) the CTF expression (Figure 4C).

### 3.6. Subcellular Localization of PS-Like Protein in Different Stages of T. cruzi

The cellular localization of the *T. cruzi* PS-like protein was determined by fluorescence microscopy using the anti-EP8 sera (Figure 5). Differences in intensity and fluorescence patterns were observed comparing pre-immune serum and anti-EP8 sera immunoreactivity (Appendix A). Merged images of EP8 staining showed a punctate staining pattern of the PS-like protein that localized to the plasma membrane and was suggestive of distinct nanodomains, especially in the area of the flagellum. Epimastigotes and amastigotes forms showed a stronger staining signal for the PS-like protein in the anterior region near the kinetoplast and flagellum (Figure 5A,C), whereas trypomastigotes showed a signal distributed along the body and in regions with an undulated membrane (Figure 5B). In epimastigotes and amastigotes, a portion of the protein was in the proximity of the flagellar pocket region near the emerging flagellum and kinetoplast (Figure 5A,C).

To analyze the influence of starvation and nutritional stress in the PS-like fluorescence distribution, epimastigotes were maintained in serum-free medium. Under this condition, an increased fluorescence signal was observed specifically in the regions of the flagellum and flagellar pocket region near the kinetoplast when compared to control groups that were cultivated with 10% FBS (Figure 5D,E). As an indication of the nutritional stress-induced by serum deprivation, an increase in the number of autophagic vacuoles in epimastigotes was observed by MDC labeling (Appendix A).

For localization of EP8 to intracellular organelles, epimastigotes were also evaluated using fluorescent trackers directed to the endoplasmic reticulum (ER), flagellar pocket, and acid vacuoles. As expected, EP8 fluorescence was mainly localized in the anterior region of the parasite (Figure 5F–H). Multichannel intensity plots related to the longitudinal axis of the parasite revealed a peak of EP8 fluorescence in the parasitic anterior region that was occurred after the fluorescence peak representing the kinetoplast. A partial correlation was observed between the PS-like fluorescence pattern and the ER, as revealed using DiOC_6_, that was distributed throughout the epimastigote (Figure 5F). In contrast, the distribution of BODIPY FL–pepstatin A, a broad aspartyl protease inhibitor, co-localized with an aspartic protease present in acidic vacuoles, both lysosomes and reservosomes, which were distributed in the posterior region of the parasites (Figure 5G).

To label the flagellar pocket, parasites were incubated with Concanavalin A-FITC (ConA-FITC) at 4 °C [38]. A strong signal was evident in the anterior region, but a fluorescence signal was also noticeable on the parasite surface. Near the kinetoplast, the localization of EP8 fluorescence correlated with the flagellar pocket tracer. An overlay of EP8 signal was partially noted against the ER probe with a greater correlation to the flagellar pocket and Cathepsin D fluorescent trackers (Figure 5F–H).

## 4. Discussion

Improvements in our understanding of the biochemistry, cellular composition, and molecular biology of *T. cruzi* can identify essential proteins and their processes that should contribute to the development of new therapies against its infections. This is especially important since the current treatments for Chagas disease include medications that display toxic side effects as well as considerable drug resistance [7], and there is little possibility of vaccine development. As the interface between a cell and its environment, the plasma membrane has a fundamental role in the transfer of material to and from the cytoplasm that also contributes to the detection and propagation of external signals [48]. Therefore, we have focused on identifying essential enzymes in the plasma membrane that could serve as targets for new therapies.

Our group previously reported the identification of an aspartyl protease activity that could be differentially distributed to a membrane-enriched fraction separate from the soluble fraction of whole parasite extracts [22]. At the time, the biochemical isolation of the enzyme responsible for the activity, at a level necessary for identification, proved to be exceedingly challenging. We chose an analysis of the *T. cruzi* genomic data as an alternative approach using sequence searches based on homologies. Using sequences of aspartyl proteases residing in the plasma membrane revealed a single sequence corresponding to a PS-like protein in *T. cruzi*. The analysis showed similarities along the primary sequence and, importantly, in highly conserved domains. Based on our previous observations [22], we hypothesized that the activity of the membrane-associated aspartic protease could initially be attributed to the PS-like protein.

Initially, a microarray of peptides was synthesized using the Spot-synthesis technique to identify linear B-cell epitopes in the putative PS-like *T. cruzi* protease that are recognized by patient antibodies. Ten epitopes were readily defined (Table 1), confirming the likelihood that the predicted *T. cruzi* PS-like protein is expressed. Molecular modeling of the sequences revealed nine TM domains (Figure 5), which was similar to the mammalian model presented [49].

To test our hypothesis that the membrane-associated aspartyl protease activity could be attributed to the presence of the *T. cruzi* PS-like protein, immunoblotting was performed with rabbit anti-EP8 epitope peptide. The estimated size of PS-like holoprotein is 41 kDa, which correlates with one of the bands detected by Western blot. Two smaller bands, one of 24 kDa and another at 32 kDa, are compatible with C-terminal fragments (CTF), which suggests that the enzyme can suffer endoproteolytic cleavage as observed for the human ortholog [23]. The presence of two CTFs is related to an alternative cleavage pathway promoted by caspases during apoptosis [50]. The induction of apoptosis could, in part, explain the increased expression of CTF observed under serum-free conditions and compound treated parasites. Additionally, the different electrophoretic mobilities of the CTFs could reflect differences in phosphorylation [51]. Cleaved and uncleaved PS was found to bind pepstatin-A [52], the identified bands of 24 and 32 kDa bound to pesptatin-A (Appendix A), but higher molecular weight bands were not found, probably due to low concentration of the uncleaved form. Although it is known that GSC can be dissociated with Triton X-100 (1%), the higher molecular mass band found (~76 kDa), could be attributed to the retention of PS-like protein with other proteins that would be consistent with its presence in a complex, such as NIC, Aph-1, and Pen-2, since some fractions can remain physically associated in high-molecular-weight complexes that are often metabolically stable [23,53].

Presenilin undergoes autoproteolytic cleavage into two subunits, N-terminal fraction and C-terminal fraction. These subunits are integral components of active γ-secretase complex and carry two important aspartyl residues in the active site [44]. Western blot with anti-EP8 sera showed an intense band at ~23 kDa correspondent to CTF. This molecular weight suggests that endoproteolytic cleavage site is present in the hydrophilic loop between transmembrane 6 and 7. The treatment of epimastigotes with GSI reduced the holoprotein expression but increased CTF. GSI drugs (DAPT and XXI/E) are known to inhibit Aβ aggregation in Alzheimer’s disease [54]. The exact mechanism of these noncompetitive proteolytic inhibitors is unknown, but DAPT appears to bind to the PS1-CTF [55]. This event was shown to increase PS1 levels in cellular and animal models [56] and stabilize the interactions between PS-CTF in the complex with APH-1/nicastrin and PS1-NTF/PEN-2 [57].

Expression of PS-CTF is low in control cultures but increased significantly in parasites incubated in serum-free medium, as well as the number of autophagic vacuoles labeled with MDC. This event correlates with previous reports that pointed to the importance of presenilins in autophagy [58,59]. Autophagy is a process that involves the degradation of cellular contents in autophagosomes and lysosomes, which is important for energy balance in response to nutritional stress. It is an important mechanism to remove misfolded or aggregated proteins, degradation of defective organelles, remove pathogens, and to maintain cytosolic amino acid pool [59]. Presenilin-1 is a key component required for autophagy and lysosomal proteolysis. It is essential for v-ATPase targeting lysosomes and its acidification [58]. PS1 Knock-out (KO) cells had a delay in the clearance of proteolytic products from autophagic vacuoles [58]. Similar features were observed in PS-1 KO blastocysts, which had a decrease in the proteolysis of long-lived proteins and increased autophagosomes. In addition, cells with a low expression of PS1 had diminished lysosomal acidification due to a dysfunction in glycosylation and the targeting of the v-ATPase VOa1 subunit [60]. With serum deprivation, the PS1 KO displayed a rapidly induced apoptosis of brain endothelial cells [61]. These lines of evidence indicate that PS1 has an essential role in the processes of autophagy and apoptosis. In *T. cruzi*, serum deprivation leads to an upregulation of *T. cruzi* PS-like protein expression that appeared linked to responses to cellular stress.

The subcellular localization of *T. cruzi* PS-like protein revealed an apparent surface concentration of signal for the protease that was observed in the anterior region of amastigotes and near the kinetoplast in epimastigotes. This corresponds to the flagellar pocket, an area of high activity in trypanosomatids that is important for parasite nutrition and other cellular processes such as cell polarity, morphogenesis, and replication [62]. It is demarcated by a small invagination of the plasma membrane where the flagellum exits the cytoplasm. The punctate intracellular signals, together with its fractionation to membranes, suggesting that the PS-like protein could also be localized to intracellular membrane-bounded structures, as observed using specific fluorescent markers (Figure 5) for the ER and acidic vesicles. It also could be associated with the Golgi complex. In other cellular systems, the integration of these organelles with PS/GSC is responsible for aspects of the secretory pathways [63,64]. In *T. cruzi* and other trypanosomatids, the secretory pathway involves the ER and Golgi complex to the flagellar pocket, which is the main site of exocytosis and endocytosis. Together, they are part of a multi-organelle complex that has also been implicated in cell polarity and cellular division [65].

While there is extensive information on the physiological role of this class of aspartyl proteases in mammalian cells, there is little knowledge of GSC's role in parasites. Its localization to the parasite's flagellar pocket may indicate that it is involved in a compartment of the cell where an intense endocytic/secretory activity of proteins occurs [48]. In eukaryotic cells, the activity of PS/GSC has been associated with intracellular trafficking and recycling of endosomal soluble proteins and membrane-associated receptors, such as transferrin receptor, through the endocytic recycling compartment [66]. In *T. cruzi*, transferrin receptors are expressed and localized in the flagellar pocket of epimastigotes and amastigotes. After transferrin binding, endosomes are delivered to reservosomes [67]. As amastigotes and epimastigotes are replicative forms, it would be expected that they would have a larger demand for molecules to sustain proliferation [68].

To evaluate the possible co-localization of PS-like within organelles like lysosomes, we incubated cells with BODIPY-FL-pepstatin A. The merged signals in immunofluorescence demonstrated that a sensitive pepstatin A aspartic protease is present in acidic vacuoles. However, it remains to be determined if this signal was exclusively due to the presence of *T. cruzi* PS-like protein or additional aspartic proteases enzyme families. The genome of *T. cruzi* predicts two additional aspartyl proteases, a Ddi-1 aspartic protease and a membrane signal peptide peptidase (SSP). The first belongs to the A2 family and is a soluble protein [66,69], while SPP is involved in ER quality control and signal peptide degradation from proteins to be exported [70]. The physicochemical properties and primary structure of the three proteinases are sufficiently different that no cross-reactive epitopes were identified (data not shown). Independently of this fact, the presence of PS-like in these organelles was also confirmed with the specific sera anti-EP8 (Figure 5G).

A protein interactome was predicted by a bioinformatic analysis to identify potential interacting partners of the *T. cruzi* PS-like protein. A putative *T. cruzi* protein and several trypanosomatids membrane proteins, including calreticulin, were identified that were consistent with a membrane association for the *T. cruzi* PS-like protein (Figure 3). *T. cruzi* calreticulin has been localized in the ER, as was the *T. cruzi* PS-like protein, and is involved in glycoprotein folding [71]. It has also been implicated in Ca^+2^ homeostasis [72]. Interestingly, human PS-1 is also involved in Ca^±^ homeostasis, in the lysosome as well as in acidification and proteolysis [73]. Predictions from the protein–protein network interaction also revealed that PS-like could possibly interact with glycogen synthase kinase, a serine/threonine kinase. In *Trypanosoma brucei*, this enzyme is involved in the regulation of transferrin endocytosis in the parasite flagellar pocket [74], a region of PS-like localization.

Phylogenetic analysis revealed distinct phylogenetic clusters of PS/PS-like proteases. For the *T. cruzi* PS-like aspartyl protease, a different subgroup was defined that contained the *T. vivax* variant, which was separate from the other protozoan subclasses, including *Leishmania* sp. and *T. brucei*. Nevertheless, the analysis indicates that the motifs crucial for γ-secretase assembly and activity are well conserved among distant evolutionary species. In the aspartic proteases from the A22 family, conserved domains of the Asp residue in the active site located in the N-terminal fragment is preceded by a Tyr residue. In addition, in the C-terminal fragment, the active site is composed of the sequence Gly-X-Gly-Asp-Phe, where X may be a Leu or a Phe residue [75]. The sequence known as the PALP domain (Pro-Ala-Leu-Pro) is well conserved within PS proteins and was identified in the *T. cruzi* protein at the TMD9 (Figure 2), similar to the human PS, which indicates that this aspartyl protease most likely belongs in the A22 family [75]. Together with the catalytic motifs GLGD and YD, the PALP domain determined the conformation of the active site [39,76,77] and was identified in all *T. cruzi* sequences analyzed (data not shown).

An identified domain in the *T. cruzi* PS-like protein was the TMD8 AxxxAxxxG sequence that also exists in human PS. This region appears to be involved in protein–protein interactions within human cells [78], which has been suggested to function in the formation and stabilization of the GSC [39].

## 5. Conclusions

This is the first description of the cellular localization of a PS-like aspartyl enzyme ortholog in *T. cruzi*. While the precise function of the *T. cruzi* PS-like enzyme is yet defined, our results show that it is expressed and upregulated under serum deprivation. Sequence analysis and epitope mapping defined it as a multi-pass membrane protein. A comparative structural analysis revealed that several motifs in the TMDs of known PSs are conserved in the *T. cruzi* PS-like protein, which suggests there could be some function conservation. Combined with its subcellular localization, we anticipate a function in secretion and/or endocytic trafficking along with autophagy. In mammalian cells, PS activates a variety of class I membrane proteins suggesting that cognate target proteins may exist in *T. cruzi*, although differences are expected as the phylogenetic analysis showed that the *T. cruzi* and vertebrate homologs were separated into divergent clades. An intensive investigation of the spatial and temporal expression patterns of the *T. cruzi* PS-like gene, combined with the use of new knock-out techniques, should help reveal its function in the host-parasite relationship and assist in the development of novel treatments that target its enzymatic activity.

## Figures and Tables

**Figure 1 biomolecules-10-01564-f001:**
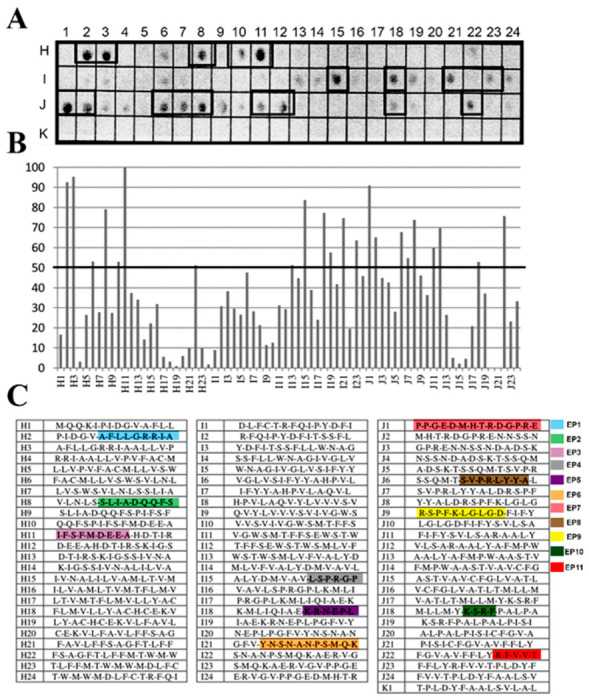
Spot synthesis analysis for linear B-cell epitopes in *T. cruzi* PS-like protein. A library of 14-mer peptides that sequentially represents the coding sequence of *T. cruzi* PS-like protein with a 9 amino acid overlap was synthesized directly onto a cellulose membrane followed by probing with a pool of Chagasic patient sera (*n* = 8). (**A**) Image of the chemiluminescent signal from bound human IgG antibodies. Peptides containing epitope sequences are identified by boxes. (**B**) Graph of the signal intensities normalized to the maximum and minimum signals from the positive and negative controls, respectively. Epitopes were identified within consecutive peptides with intensity levels above 50%. (**C**) Table of the individual 14-mer peptides and their locations on the membrane.

**Figure 2 biomolecules-10-01564-f002:**
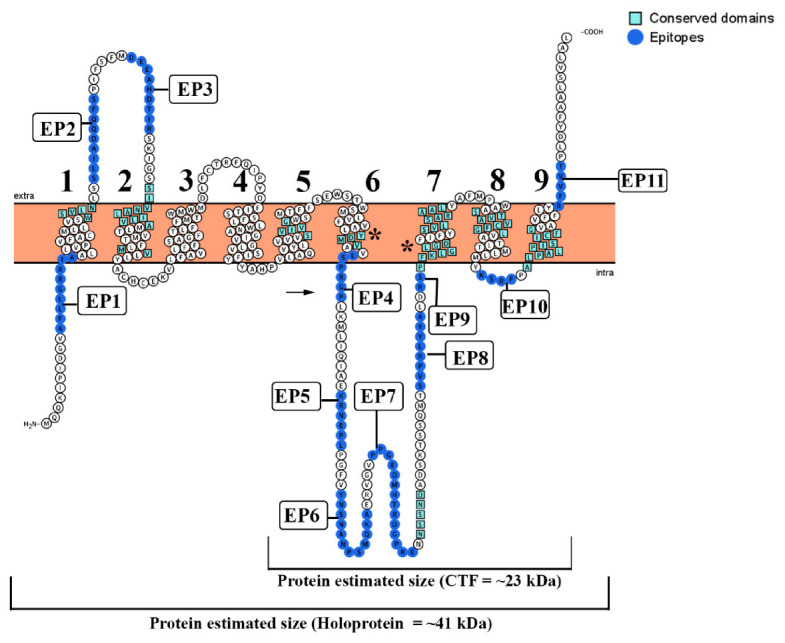
Modeling of *T. cruzi* PS as a multi-pass transmembrane protein. The model is based on the predictive results obtained using TopCons (https://topcons.net/pred), and layout was generated using Protter (http://wlab.ethz.ch/protter/). The model with 9 transmembrane domains reveals all 11 identified epitopes, probable sites of endoproteolytic cleavage (arrow), and the localization of the typical domains listed in Table 2. (*) Highly conserved domain in the PS-like catalytic pocket (YD and GLGD).

**Figure 3 biomolecules-10-01564-f003:**
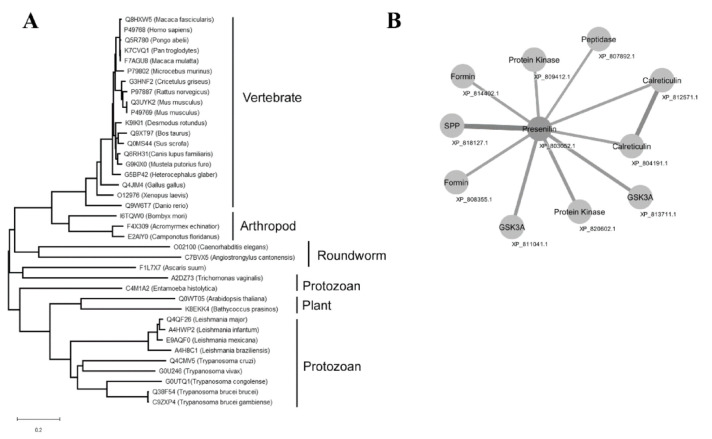
Phylogenetic relationship of PS and PS-like proteins and protein–protein interaction network of *T. cruzi* PS-like. (**A**) Amino acid sequences obtained from Uniprot server (http://www.uniprot.org, search criteria name: “PS” and length: [300 to 800]) were aligned with Clustal W and the phylogenetic tree was constructed with the sequences aligned by the neighbor-joining algorithm using a CLUSTAL W in MEGA software. Family members are grouped according to their relationship to human SPP/SPPL orthologs. *Acromyrmex echinatior* (F4 × 309), *Angiostrongylus cantonensis* (C7BVX5), *Arabidopsis thaliana* (Q0WT05), *Ascaris suum* (F1L7 × 7), *Bathycoccus prasinos* (K8EKK4), *Bombyx mori* (I6TQW0), *Bos taurus* (Q9XT97), *Caenorhabditis elegans* (O02100), *Camponotus floridanus* (E2AIY0), *Canis familiaris* (Q6RH31), *Cricetulus griséus* (G3HNF2), *Danio rerio* (Q9W6T7), *Desmodus rotundus* (K9IKI1), *Entamoeba histolytica* (C4M1A2), *Gallus gallus* (Q4JIM4), *Heterocephalus glaber* (G5BP42), *Homo sapiens* (P49768), *Macaca fascicularis* (Q8HXW5), *Macaca mulatta* (F7AGU8), *Microcebus murinus* (P79802), *Mus musculus* (P49769), *Mus musculus* (Q3UYK2), *Mustela putorius furo* (G9KIX0), *Pan troglodytes* (K7CVQ1), *Pongo abelii* (Q5R780), *Rattus norvegicus* (P97887), *Sus scrofa* (Q0MS44), *Xenopus laevis* (O12976), *Trichomonas vaginalis* (A2DZ73), *Leishmania mexicana* (E9AQF0), *Leishmania major* (Q4QF26), *Leishmania infantum* (A4HWP2), *Leishmania braziliensis* (A4H8C1), *Trypanosoma cruzi* (Q4E0Z2), *Trypanosoma brucei* brucei (Q38F54), *Trypanosoma brucei gambiense* (C9ZXP4), *Trypanosoma congolense* (G0UTQ1), *Trypanosoma vivax* (G0U246). (**B**) Network analysis of *T. cruzi* PS-like protein interactions curated in STRING: Calreticulin, signal peptide peptidase (SPP), peptidase, formin, protein kinase, and glycogen synthase kinase 3A (GSK3A). Edge line thickness represents the strength of data support.

**Figure 4 biomolecules-10-01564-f004:**
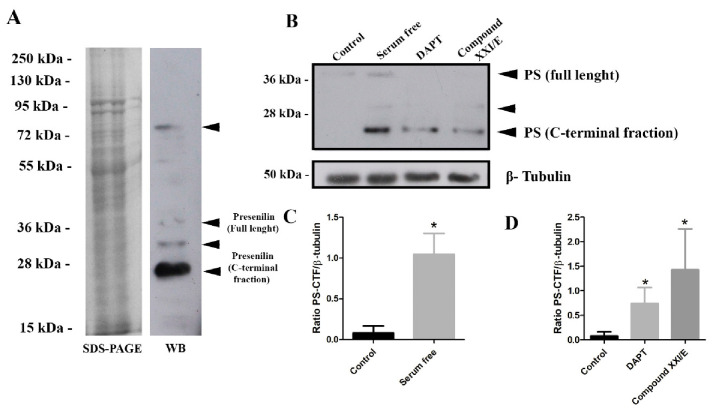
Performance of the anti-EP8 rabbit polyclonal serum for detection of the *T. cruzi* PS-like protein. (**A**) SDS-PAGE of CL strain epimastigote whole extract (20 µg) and a corresponding Western blot probed with anti-EP8 serum. Four main bands were identified (arrows). (**B**) Western blot of whole-cell extracts (20 µg) from epimastigotes under serum deprivation conditions and after treatment with gamma-secretase inhibitors, DAPT (100 µM) and Compound XXI/E (200 µM) after 24 h with polyclonal anti-EP8 serum or anti-β-tubulin antibodies (internal control). Densitometry analysis of the Western blots presented as the ratio of *T. cruzi* PS-CTF and β-tubulin under nutritional stress (**C**) and treated with DAPT (100 µM) or compound XXI/E (200 µM) (**D**). Data represent the mean and standard deviation from at least three independent experiments. * Significant difference using *t*-test (*p* < 0.05).

**Figure 5 biomolecules-10-01564-f005:**
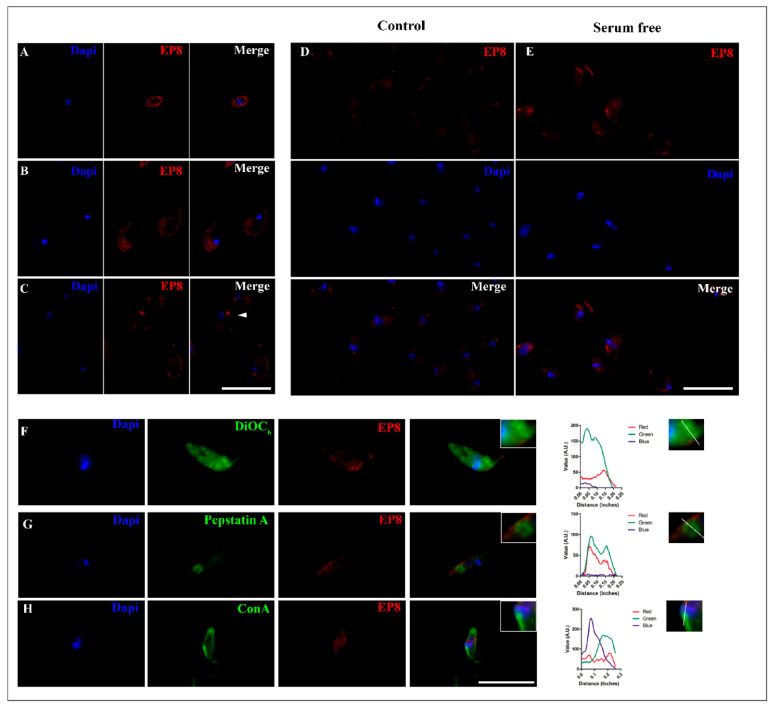
Immunofluorescent subcellular localization of the *T. cruzi* PS-like protein. A single z-plane image of a representative parasite of different forms of *T. cruzi*, (**A**) amastigote (**B**) trypomastigote and (**C**) epimastigote immunolabeled with anti-EP8 (red) and DAPI to label nuclei and kinetoplast (blue). White arrowhead shows an immunolabeled concentration signal near the flagellar pocket. Increased signal of PS-like protein in epimastigotes from control conditions in 10% FBS (Control; (**D**)) to serum deprivation for 24 h (**E**). Cytolocalization of EP8 with the ER stained with DiOC6 (**F**), intracellular vesicles marked with Cathepsin B (**G**) and the flagellar pocket (**H**). Fluorescence intensity plot of each channel (Red, Green, and Blue) were performed, tracing a line from the posterior to the anterior region of the parasite for each organelle dye. Scale bar = 10 µm.

**Table 1 biomolecules-10-01564-t001:** Epitopes mapped in *Trypanosoma cruzi* PS-like protein (Q4CMV5) using a pool of sera from patients with chronic Chagas disease.

Epitope Code	Epitope Sequence	Residue Position
EP1	AFLLGRRIA	11–19
EP2	SLIADQQFS	41–49
EP3	IFSFMDEEA	51–58
EP4	ALYDMVAVLSPRGP	199–204
EP5	KRNEPL	214–219
EP6	YNSNANPSMQKA	224–235
EP7	PPGEDMHTRDGPRE	241–254
EP8	SVPRLYYA	271–278
EP9	RSPFKLGLGD	281–290
EP10	KSRF	331–334
EP11	RFVVT	355–359

**Table 2 biomolecules-10-01564-t002:** Comparison of amino acid motifs in human PS and the *T. cruzi* orthologue.

TMD	*T. cruzi* PS-Like	aa Position *	Human PS	AA Position *
1st	WSVLN	34–38	ATIKS	98–102
2nd	SIVNALILVA	70–79	SILNAAIMIS	132–141
2nd	MV	88–89	MV	93–94
5th	SVIVG	168–172	SALMA	230–234
6th	YD	193–194	YD	256–257
6th	MV	195–196	MV	298–299
-	NSSND	256–260	AQRDS	342–346
-	-	-	SSILA	366–370
7th	PFKLGLGD	283–290	GVKLGLGD	378–385
7th	SVLSARAAL	295–303	SVLVGKASA	390–398
8th	ASTVAVCFG	312–320	ACFVAILIG	409–417
9th	PALP	339–342	PALP	433–436
9th	ALPALPISICFG	336–347	ALPALPISITFG	431–442

* Amino acid position in PS primary structure; Grey highlight denotes amino acids differences between *T. cruzi* and human segments.

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
