# Peer review of "Trypanosoma cruzi Presenilin-Like Transmembrane Aspartyl Protease: Characterization and Cellular Localization"

_biomolecules, 2020, doi:10.3390/biom10111564_

Round 1
Reviewer 1 Report
The work by S. G. De-Simone et al has been very well performed and will be useful for the scientific community. Just some minor changes should be made before publication:
-In reference 1 says "World Heal. Organ." but should say "World Health Organization".
-In reference 12 says "Benznidazole and posaconazole in eliminating parasites in asymptomatic T. cruzi carriers: The STOP-623 CHAGASt." but should say "Benznidazole and posaconazole in eliminating parasites in asymptomatic T. cruzi carriers: The STOP-623 CHAGAS Trial."
-In reference 13 says "Annu, Rev, Biochem," but should say "Annu. Rev. Biochem."
-In reference 37 says "Analyt. Biochem." but should say "Anal. Biochem."
Author Response
We appreciate efforts on the part of the reviewers to improve the presentation of our research and their careful attention to even the smallest details. Below, we have included our response in italics to each suggestion, comment and criticism. All changes in the revised manuscript are highlighted in yellow to facilitate its review.
Reviewer 1:
The work by S. G. De-Simone et al has been very well performed and will be useful for the scientific community. Just some minor changes should be made before publication:
-In reference 1 says "World Heal. Organ." but should say "World Health Organization".
-In reference 12 says "Benznidazole and posaconazole in eliminating parasites in asymptomatic T. cruzi carriers: The STOP-623 CHAGASt." but should say "Benznidazole and posaconazole in eliminating parasites in asymptomatic T. cruzi carriers: The STOP-623 CHAGAS Trial."
-In reference 13 says "Annu, Rev, Biochem," but should say "Annu. Rev. Biochem."
-In reference 37 says "Analyt. Biochem." but should say "Anal. Biochem."
Author response: Thank you for the comments on our work. It is very satisfying to receive positive feedback from someone whose role is carefully to critique a study to identify issues prior to publication. The errors detected in each of the references have been corrected.
Reviewer 2 Report
With the motivation to more precisely dissect the biology and metabolism of parasites such as Chagas disease-causing Trypanosoma cruzi in order to ultimately identify novel potential drug targets, the manuscript submitted by Lechuga et al. provides a first characterization of the Trypanosoma cruzi orthologue of vertebrate presenilin, an intramembrane-cleaving aspartyl protease with a pivotal function in the proteolytic processing of type I membrane proteins. While the topic is undoubtedly of high relevance, it is my opinion that this study needs to be developed further before publication.
First, the manuscript has to be revised as – in its current form – it is inconsistent and self-contradictory. An example is the overall topology of the presenilin gene characterized in this study. Using prediction algorithms, the authors propose a 9-TMD topology as depicted in Fig. 4 (an obviously expected finding), but several times within this very same manuscript the authors appear to refer to a 7-TMD topology (e.g. lines 456ff, 549 ff). This should be changed to one consistent narrative.
The authors should be more careful in describing novelty aspects in this manuscript. Contradicting a statement in line 72, the sheer presence of a presenilin-like gene in T. cruzi has been known for a while (corresponding entries are found in MEROPS, Uniprot, and such insights have been reported already; e.g. statement in Alvarez et al. BBA 2011).
Several times throughout the manuscript the authors refer to their previous line of work into aspartyl proteases of T. cruzi. At that time, they identified two distinct Pepstatin A-sensitive protease activities in T. cruzi extracts – one potentially residing within a membranous compartment, triggering their interest in GxGD-type aspartyl proteases. Now, with their newly generated antiserum, it would be interesting to see whether their protease-enriched fraction at that time contains the PS-like protein. Of note, T. cruzi likely also harbors a SPP orthologue (Q4DUR7), a fact that deserves to mentioned here as well in light of SPPs and presenilins close relationship.
Regarding the epitope mapping presented in section 3.1:
- Main text and Fig 1 legend are inconsistent: line 252/253 “… ten epitopes showed signal intensities above 30% that was considered cutoff…” vs. line 261 “epitopes were defined by intensity levels above 50%”… the bold line in Fig 1b is in fact at 50%, but more than 10 peptides displayed signal intensities above that. It might be useful to more clearly differentiate between the terms “peptides” and “epitopes” here
- It would be helpful to highlight the identified epitopes (EP1 to 10) summarized in Tab. 10 also in the corresponding peptide sequences summarized in Fig. 1C (colorcoding or the like).
Regarding section 3.2:
- line 278f: Given that this manuscript was submitted for publication, I consider such a statement regarding preliminary data inadequate. It would be good to present such data in a supplemtary figure
- main text (l. 283) and Fig 2 legend (l. 293) refer to 4 bands that are detected by anti-EP7 antiserum and supposedly marked by arrows in Fig. 2A, but only two arrows are shown
- the data presented in Fig. 2 is not entirely sufficient to prove specificity of anti-PE7 for T. cruzi presenilin (this similarly applies to the IF stainings in Fig. 3!). Though described to work less well in immunofluorescence, how does anti-EP8 perform in Western? Does it similarly detect the putative PS full-length and CTF bands? Can binding of anti-EP7 be competed away with the blocking peptide? It would be most convincing to clone and overexpress T. cruzi PS heterologously e.g. in HEK cells to prove antibody specificity.
- the Western blot in Fig. 2B suggests that PS CTF is upregulated by serum deprivation. This would actually mean that PS expression is upregulated and the holoprotein is subsequently almost quantitatively turned over into CTF and NTF. Can the upregulation of PS expression be corroborated e.g. RT-PCR data?
- how can the PS CTF accumulation observed after DAPT treatment be explained (considering such an effect has not been reported before for human PS)
- in the discussion (referring to Fig 2, I assume), two separate CTFs (24 and 32 kDa) are mentioned – how can two distinct PS CTF species be explained?
Section 3.6/STRING analysis:
The benefit from this analysis is not evident to the reviewer. There is essentially no experimental data on T. cruzi PS interactions and the data underlying the STRING plot are based on textmining or interactions of homologues of the respective proteins in other species (scored with low confidence level). In line with this, the corresponding section of the discussion (lines 532f) should be rephrased.
Line 562:
In the field of intramembrane proteases, the term “type I/class I membrane protein” generally refers to the topology of single-pass membrane proteins with Noutside-Cinside topology. “Multi-pass membrane protein” might be a good replacement here.
Minor suggestion: In regards to readability and storytelling, the manuscript could benefit from changes in structure. It would be helpful to the ready for instance, if the in silico characterization of PS and its topology (i.e. 3.4 – 3.6) are described first in the results section, followed by the experimental work (3.1 to 3.3).
Author Response
We appreciate efforts on the part of the reviewers to improve the presentation of our research and their careful attention to even the smallest details. Below, we have included our response in italics to each suggestion, comment, and criticism. All changes in the revised manuscript are highlighted in yellow to facilitate its review.
Reviewer 2:
With the motivation to more precisely dissect the biology and metabolism of parasites such as Chagas disease-causing Trypanosoma cruzi in order to ultimately identify novel potential drug targets, the manuscript submitted by Lechuga et al. provides a first characterization of the Trypanosoma cruzi orthologue of vertebrate presenilin, an intramembrane-cleaving aspartyl protease with a pivotal function in the proteolytic processing of type I membrane proteins. While the topic is undoubted of high relevance, it is my opinion that this study needs to be developed further before publication.
1) First, the manuscript has to be revised as – in its current form – it is inconsistent and self-contradictory. An example is the overall topology of the presenilin gene characterized in this study. Using prediction algorithms, the authors propose a 9-TMD topology as depicted in Fig. 4 (an obviously expected finding), but several times within this very same manuscript the authors appear to refer to a 7-TMD topology (e.g. lines 456ff, 549 ff). This should be changed to one consistent narrative.
Author response:- This inconsistency has been corrected in the manuscript-
“Molecular modeling of the sequences revealed nine TM domains (Figure 2), which was similar to the mammalian model presented [49]”.
2) Using predicted algoritm...(an obvious expected finding).
Author response: We respectfully disagree with the referee, but we do not believe that it is obvious that all eukaryotic cell PSs with 9 TM without empirical data. This is because the software utilized in our study (ff-29-34) presented multiple models containing 7, 8, and 9 TM for the T. cruzi PS. It was our determination of the epitopes that resolved which model was mostly correctly based on the fact that an epitope cannot exist within the middle of a bilayer membrane, as some models suggested. Ultimately, the best model that localized all epitopes in antibody accessible positions and maintained the other domains that are similar to human PS in their respective regions was the 9 TM model. Further, since the length of the proteins is variable, different numbers of TM domains may occur. For a hookworm PS-like protein, our studies have shown that it has 7 TM and not 9. This aspartyl protease is about 20% smaller than T. cruzi (357 aa), which is also smaller than the human version (467 aa). Finally, in humans, there are two PS copies and in T. cruzi there is only a single gene predicted, so organisms are biologically different and only experimentation can demonstrate these similarities and differences.
3) The authors should be more careful in describing novelty aspects in this manuscript. Contradicting a statement inline 72, the sheer presence of a presenilin-like gene in T. cruzi has been known for a while (corresponding entries are found in MEROPS, Uniprot, and such insights have been reported already; e.g. statement in Alvarez et al. BBA 2011).
Author response:- We have been specific on the novelty of our research to indicate that it is the first description of its subcellular localization. The reference pointed out by the referee “Vania E. Alvarez et al BBA 1824 (2012) 195-206”, lists in table 1 all existing proteases and those supposed to exist in T. cruzi at that time, based on homology with T. brucei proteins. No further information on putative protein is given throughout the paper. It is worth noting that the structural homology between presenilins is 20-30% and it is, therefore, difficult to use this information with the certainty of the protein's function.
4) Several times throughout the manuscript the authors refer to their previous line of work into aspartyl proteases of T. cruzi. At that time, they identified two distinct Pepstatin A-sensitive protease activities in T. cruzi extracts – one potentially residing within a membranous compartment, triggering their interest in GxGD-type aspartyl proteases. Now, with their newly generated antiserum, it would be interesting to see whether their protease-enriched fraction at that time contains the PS-like protein.
Author response:- These experiments were performed, and they confirmed the presence of the 25 KDa band in hydrophobic membrane preparations. However, we believe the data do not warrant inclusion as they confirmed our previous data (Pinho et al., 2007) and would have made the article longer.
6) Of note, T. cruzi likely also harbors an SPP orthologue (Q4DUR7), a fact that deserves to mentioned here as well in light of SPPs and presenilins close relationship.
Author response: -In fact, all comparative studies with possible SPP of T. cruzi have been carried out and the protein Q4DUR7 pointed out by the reviewer is structurally different, whether due to the way it is oriented (Nâ–ºC) on the cell membrane and the presence of the epitopes described in this study for PS. However, we agree with the referee, the importance of the information and insert a sentence stating:
“The genome of T. cruzi predicts two additional aspartyl proteases, a Ddi-1 aspartic protease and a membrane signal peptide peptidase (SSP). The first belongs to the A2 family and is a soluble protein [66,69] while SPP is involved in ER quality control and signal peptide degradation from proteins to be exported [70]. The physicochemical properties and primary structure of the three proteinases are sufficiently different that no cross-reactive epitopes were identified (data not shown). Independently of this fact, the presence of PS-like in these organelles was also confirmed with the specific sera anti-EP7 (Figure 5 G).”
7) Regarding the epitope mapping presented in section 3.1: Main text and Fig 1 legend are inconsistent: line 252/253 “… ten epitopes showed signal intensities above 30% that was considered cutoff…” vs. line 261 “epitopes were defined by intensity levels above 50%”… the bold line in Fig 1b is in fact at 50%, but more than 10 peptides displayed signal intensities above that. It might be useful to more clearly differentiate between the terms “peptides” and “epitopes” here
Author response: -Thank you for identifying this inconsistency. This has been corrected to read:
Text: “From the 73 peptides in the library, twenty peptides showed signal intensities above 50% that was considered above the cutoff for a positive reaction (summarized in Table 1).”
Legend: “Epitopes were identified within consecutive peptides with intensity levels above 50%.”
8) It would be helpful to highlight the identified epitopes (EP1 to 10) summarized in Tab. 10 also in the corresponding peptide sequences summarized in Fig. 1C (color coding or the like).
Author response: We appreciate the suggestion and have modified Fig 1 to improve the presentation of the epitopes.
9) Regarding section 3.2:-line 278f: Given that this manuscript was submitted for publication, I consider such a statement regarding preliminary data inadequate. It would be good to present such data in a supplementary figure-
Author response:- We understand the referee’s concern over a reference to data not shown. Unfortunately, our primary image data did not meet the standards of the journal. As it was not the only data used to choose E7 over EP8, we felt that we could make such a reference. Based on your comment, we have performed a peptide ELISA and a competition assay to support our conclusion on the higher performance of E7. The results are in the supplemental material and an additional sentence was added in the results section.
“Additionally, peptide ELISAs confirmed the higher performance of the anti-EP7 sera as well as a competition assay that showed a sharp inhibition activity by preincubation with an increasing concentration of EP7 peptide. An inhibition of 50% was obtained with 20 µM of the peptide (Figure S2). Based on these results, the rabbit antisera anti-EP7 was chosen for further characterization of the T. cruzi PS-like protein”
10)- main text (l. 283) and Fig 2 legend (l. 293) refer to 4 bands that are detected by anti-EP7 antiserum and supposedly marked by arrows in Fig. 2A, but only two arrows are shown.
Author response:- The reviewer is correct, all four bands should have been indicated by arrows instead of only highlighting the full-length and CTF of PS. A new figure depicting the other bands has been included.
11)- the data presented in Fig. 2 is not entirely sufficient to prove the specificity of anti-PE7 for T. cruzi presenilin (this similarly applies to the IF stainings in Fig. 3!). Though described to work less well in immunofluorescence, how does anti-EP8 perform in Western? Does it similarly detect the putative PS full-length and CTF bands? Can binding of anti-EP7 compete away with the blocking peptide? It would be most convincing to clone and overexpress T. cruzi PS heterologously e.g. in HEK cells to prove antibody specificity.-
Author response:- The decision to generate both anti-EP7 and anti-EP8 was to improve the chance of obtaining at least a single, high-quality anti-PS-like protein reagent. After our initial evaluations of their performance and specificity, we chose to focus entirely on anti-EP7 and did not test EP8 by western blot. While the expression of the T. cruzi PS-like protein would be an excellent confirmatory approach to evaluate specificity, the reality in Brazil is that it would be expensive, laborious, and take an undeterminable amount of time that would greatly delay publication. We believe the additional competition peptide experiment, presented above. demonstrates that the anti-serum is specific to the peptide for EP7 and when combined with the restricted presence of the sequence exclusively to the PS-like coding sequence provides sufficient confidence in the supposition that the antisera are only recognizing the PS-like protein.
12) - the Western blot in Fig. 2B suggests that PS CTF is upregulated by serum deprivation. This would actually mean that PS expression is upregulated and the holoprotein is subsequently almost quantitatively turned over into CTF and NTF. Can the upregulation of PS expression be corroborated e.g. RT-PCR data?
Author response: RT-PCR is not an effective method as the control of gene expression in T. cruzi is not regulated by the production of RNA, rather by trans-splicing and post-transcriptional modifications.
13)- how can the PS CTF accumulation observed after DAPT treatment be explained (considering such an effect has not been reported before for human PS)-
Author response: Indeed, it was an unexpected result and we cannot fully explain the observation. However, despite the level of similarity in the conserved domains and topology of T. cruzi PS-like protein to human PS, there is only a 30% identity overall that can explain the absence of a similar effect with human PS.
14) - in the discussion (referring to Fig 2, I assume), two separate CTFs (24 and 32 kDa) are mentioned – how can two distinct PS CTF species be explained? -
Author response: During apoptosis, caspases appear to cleave PS1 into different molecular weight fragments that vary by electrophoretic mobility, producing PS1-CTF1&2. It is difficult to predict exactly the molecular weight of the CTFs since gel mobility could be impacted by phosphorylation. Additional information was added to the discussion section that cites two additional references:
“The presence of two CTFs is related to an alternative cleavage pathway promoted by caspases during apoptosis [50]. The induction of apoptosis could in part explain increased expression of CTF in serum-free and compound treated parasites. Additionally, different electrophoretic mobility of CTFs could be impacted by residues phosphorylation [51].” Please see the references below:
- Walter J, Grünberg J, Capell A, Pesold B, Schindzielorz A, Citron M, Mendla K, George-Hyslop PS, Multhaup G, Selkoe DJ, Haass C. Proteolytic processing of the Alzheimer disease-associated presenilin-1 generates an in vivo substrate for protein kinase C. Proc Natl Acad Sci U S A. 1997 May 13;94(10):5349-54. doi: 10.1073/pnas.94.10.5349. PMID: 9144240; PMCID: PMC24681.
- Kim TW, Pettingell WH, Jung YK, Kovacs DM, Tanzi RE. Alternative cleavage of Alzheimer-associated presenilins during apoptosis by a caspase-3 family protease. Science. 1997 Jul 18;277(5324):373-6. doi: 10.1126/science.277.5324.373. PMID: 9219695.
15) Section 3.6/STRING analysis:- The benefit of this analysis is not evident to the reviewer. There is essentially no experimental data on T. cruzi PS interactions and the data underlying the STRING plot are based on text mining or interactions of homologs of the respective proteins in other species (scored with low confidence level). In line with this, the corresponding section of the discussion (lines 532f) should be rephrased. –
Author response:-We agree with the point of view of the referee that data and text mining must be used carefully. Since little is known about this protein, we knew there was no experimental data on its interactions with other proteins. The STRING analysis is included as an opening to future experiments that aim to elucidate its function and potential as a therapeutic target. We have rephrased the paragraph to read:
“A protein interactome was predicted by a bioinformatic analysis to identify potential interacting partners of the T. cruzi PS-like protein. A putative T. cruzi protein and several trypanosomatids membrane proteins, including calreticulin, were identified that were consistent with a membrane association for the T. cruzi PS-like protein (Figure 3). T. cruzi calreticulin has been localized in the ER, as was the T. cruzi PS-like protein, and is involved in glycoprotein folding [71]. It has also been implicated in Ca+2 homeostasis [72]. Interestingly, human PS-1 is also involved in Ca± homeostasis, in the lysosome as well as in acidification and proteolysis [73]. Predictions from the protein-protein network interaction also revealed that PS-like could possibly interact with glycogen synthase kinase, a serine/threonine kinase. In Trypanosoma brucei, this enzyme is involved in the regulation of transferrin endocytosis in the parasite flagellar pocket [74], a region of PS-like localization.”
16) Line 562: In the field of intramembrane proteases, the term “type I/class I membrane protein” generally refers to the topology of single-pass membrane proteins with N outside-C inside topology. “Multi-pass membrane protein” might be a good replacement here.
Author response:-Thank you for the suggestion, the sentence was modified:
“Sequence analysis suggests that it is a multi-pass membrane protein and its subcellular localization is suggestive for a function in secretion…”
17) Minor suggestion: In regard to readability and storytelling, the manuscript could benefit from changes in structure. It would be helpful to the ready for instance, if the in silico characterization of PS and its topology (i.e. 3.4 – 3.6) are described first in the results section, followed by the experimental work (3.1 to 3.3).-
Author response:- As suggested to improve readability the structure of the results section was changed.

Round 2
Reviewer 2 Report
I thank the authors for addressing a number of my concerns. Several of these have been addressed to the extent that they can be considered resolved. In some cases this have been issues regarding text editing and data representation, but the authors have also further substantiated specificity of their antiserum with new experimental data. There are, however, some remaining issues:
New reviewer’s comment:
Final remark in the abstract: “…parasitic PS-like enzymes […] serve as biomarker for infections” – In it’s current form, I think this statement is rather far-fetched and should be omitted (or be further substantiated).
Reviewer’s previous comment:
Several times throughout the manuscript the authors refer to their previous line of work into aspartyl proteases of T. cruzi. At that time, they identified two distinct Pepstatin A-sensitive protease activities in T. cruzi extracts – one potentially residing within a membranous compartment, triggering their interest in GxGD-type aspartyl proteases. Now, with their newly generated antiserum, it would be interesting to see whether their protease-enriched fraction at that time contains the PS-like protein.
Author response:- These experiments were performed, and they confirmed the presence of the 25 KDa band in hydrophobic membrane preparations. However, we believe the data do not warrant inclusion as they confirmed our previous data (Pinho et al., 2007) and would have made the article longer.
New reviewer’s comment:
The author’s statement is relevant in regards to the Fig. 4A. In immunoblotting, the antiserum detects multiple bands, yet the nature of the individual bands remains rather speculative. The author’s statement that the 25 kDa can be confirmed in the membrane fraction is very interesting as it would further support that the antiserum specifically detects membranous PS-like protein. Yet it also raises the question as to the nature of the other bands. Are they similarly enriched in membrane fractions (expected for PS and PS fragments). Since already available, including such Western blot data shuld be included to strengthen the authors’ findings.
Reviewer’s previous comment:
Regarding the epitope mapping presented in section 3.1: Main text and Fig 1 legend are inconsistent: line 252/253 “… ten epitopes showed signal intensities above 30% that was considered cutoff…” vs. line 261 “epitopes were defined by intensity levels above 50%”… the bold line in Fig 1b is in fact at 50%, but more than 10 peptides displayed signal intensities above that. It might be useful to more clearly differentiate between the terms “peptides” and “epitopes” here
Author response: -Thank you for identifying this inconsistency. This has been corrected to read:
Text: “From the 73 peptides in the library, twenty peptides showed signal intensities above 50% that was considered above the cutoff for a positive reaction (summarized in Table 1).”
Legend: “Epitopes were identified within consecutive peptides with intensity levels above 50%.”
New reviewer’s comment:
Thank you for clarifying this a bit further. But this leads to the question why region of the peptide represented by H11 was not classified as “epitope”. After all, peptide H11 led to the strongest signal in the whole experiment.
Reviewer’s previous comment:
the data presented in Fig. 2 is not entirely sufficient to prove the specificity of anti-PE7 for T. cruzi presenilin (this similarly applies to the IF stainings in Fig. 3!). Though described to work less well in immunofluorescence, how does anti-EP8 perform in Western? Does it similarly detect the putative PS full-length and CTF bands? Can binding of anti-EP7 compete away with the blocking peptide? It would be most convincing to clone and overexpress T. cruzi PS heterologously e.g. in HEK cells to prove antibody specificity.
Author response:
The decision to generate both anti-EP7 and anti-EP8 was to improve the chance of obtaining at least a single, high-quality anti-PS-like protein reagent. After our initial evaluations of their performance and specificity, we chose to focus entirely on anti-EP7 and did not test EP8 by western blot. While the expression of the T. cruzi PS-like protein would be an excellent confirmatory approach to evaluate specificity, the reality in Brazil is that it would be expensive, laborious, and take an undeterminable amount of time that would greatly delay publication. We believe the additional competition peptide experiment, presented above. demonstrates that the anti-serum is specific to the peptide for EP7 and when combined with the restricted presence of the sequence exclusively to the PS-like coding sequence provides sufficient confidence in the supposition that the antisera are only recognizing the PS-like protein.
New reviewer’s comment:
While the competition data undoubtedly strengthen the good binding properties of anti-EP7 antiserum, these cannot rule out that this antiserum binds (an)other protein(s) in the T. cruzi whole cell extract. To rule this out, it would be critical to have a control setting with proven higher/lower expression of the PS-like protein. Blotting of membrane fractions would also be helpful here. I have similar concerns regarding the immunostaining in Fig. 5.
Author Response
We appreciate the acceptance of our manuscript for publication by Reviewer 1 and respect the additional comments by Reviewer 2. Below, we have included our newest responses in italics to address their concerns. All changes in the second revised manuscript are highlighted in teal while maintaining the yellow of the first revision to facilitate its review.
2.1) Overall reviewer comment: I thank the authors for addressing a number of my concerns. Several of these have been addressed to the extent that they can be considered resolved. In some cases, this has been issues regarding text editing and data representation, but the authors have also further substantiated the specificity of their antiserum with new experimental data. There are, however, some remaining issues:
New reviewer’s comment: Final remark in the abstract: “…parasitic PS-like enzymes […] serve as a biomarker for infections” – In its current form, I think this statement is rather far-fetched and should be omitted (or be further substantiated).
New author response: Upon reflection of the reviewer’s comment, we have changed the phrase to read:”… serve as a target for the generation of new therapeutics specific the T. cruzi.”
2.2) Reviewer’s previous comment: Several times throughout the manuscript, the authors refer to their previous line of work into aspartyl proteases of T. cruzi. At that time, they identified two distinct Pepstatin A-sensitive protease activities in T. cruzi extracts – one potentially residing within a membranous compartment, triggering their interest in GxGD-type aspartyl proteases. Now, with their newly generated antiserum, it would be interesting to see whether their protease-enriched fraction at that time contains the PS-like protein.
Previous author response:- These experiments were performed, and they confirmed the presence of the 25 KDa band in hydrophobic membrane preparations. However, we believe the data do not warrant inclusion as they confirmed our previous data (Pinho et al., 2007) and would have made the article longer.
New reviewer’s comment: The author’s statement is relevant in regards to the Fig. 4A. In immunoblotting, the antiserum detects multiple bands, yet the nature of the individual bands remains rather speculative. The author’s statement that the 25 kDa can be confirmed in the membrane fraction is very interesting; as it would further support that the antiserum specifically detects membranous PS-like protein. Yet it also raises the question as to the nature of the other bands. Are they similarly enriched in membrane fractions (expected for PS and PS fragments). Since already available, including such Western blot data should be included to strengthen the authors’ findings.
As suggested, the image was added to the supplemental material. Parasite extracts were separated into soluble and hydrophobic membrane fraction (detergent) using CHAPS and submitted to affinity chromatography using pepstatin-agarose. As a previously result Western blot was performed using anti-detergent fraction and a pool of chagasic sera. The approximately 25-kDa band was revealed using the rabbit anti-detergent fraction. In addition, to the strength that this fragment is indeed PS-like protein we performed a new assay showing epimastigote proteins adsorption to pepstatin-agarose, revealed using anti-EP7 (Now anti-EP8) sera. As expected, the ~ 25-kDa band and ~ 32-kDa band were also identified; both bands signals appeared enriched by pepstatin-agarose binding, compared to an unbound fraction. Higher bands corresponding to PS holoprotein were not identified, probably due to low concentration of the uncleaved form.
Sentences were added to the manuscript in the results and discussion section:
“T. cruzi PS-like transmembrane protein is expected to be identified in membrane preparations for that we separated a soluble and detergent fraction using CHAPS (1%) and performed a pepstatin-A affinity chromatography. Western blot using anti-detergent fraction immunized rabbit sera and a pool of sera from chagasic patients revealed some reactive bands of approximately 70, 67, 45, 43, 36, and 23 kDa (Figure S3).”
“Western blot analysis of epimastigote whole extract revealed 4 major bands, a stronger band at ~23 kDa, and others at ~32 kDa, 40 kDa and 76 kDa (Figure 4A). To test if PS-like protein bind to its specific inhibitor pepstatin A, parasite lysate was adsorbed to pepstatin A-agarose and bound proteins analyzed by western blot using anti-EP7 sera (now EP8), that identified bands of ~23 kDa and ~32 kDa (Figure S4).”
“Cleaved and uncleaved PS was found to bind pepstatin-A [52], the identified bands of 24 and 32 kDa bound to pesptatin-A (Figure S4), but higher bands were not found, probably due to low concentration of the uncleaved form.”
2.3) Reviewer’s previous comment: Regarding the epitope mapping presented in section 3.1: Main text and Fig 1 legend are inconsistent: line 252/253 “… ten epitopes showed signal intensities above 30% that was considered cutoff…” vs. line 261 “epitopes were defined by intensity levels above 50%”… the bold line in Fig 1b is in fact at 50%, but more than 10 peptides displayed signal intensities above that. It might be useful to more clearly differentiate between the terms “peptides” and “epitopes” here
Previous author response: -Thank you for identifying this inconsistency. This has been corrected to read:
Text: “From the 73 peptides in the library, twenty peptides showed signal intensities above 50% that was considered above the cutoff for a positive reaction (summarized in Table 1).”
Legend: “Epitopes were identified within consecutive peptides with intensity levels above 50%.”
New reviewer’s comment: Thank you for clarifying this a bit further. However, this leads to the question of why the region of the peptide represented by H11 was not classified as an “epitope”. After all, peptide H11 led to the strongest signal in the whole experiment.
New author Response: This forgetfulness was corrected throughout the text. Initially, this epitope had been considered to have a longer sequence (aa49-aa62), but after refinement, it was divided into 2 epitopes (now EP2 and the new EP3). Therefore, instead of 10 epitopes, there are now 11 epitopes.
2.4) Reviewer’s previous comment: The data presented in Fig. 2 is not sufficient to prove the specificity of anti-PE7 for T. cruzi presenilin (this similarly applies to the IF stainings in Fig. 3!). Though described to work less well in immunofluorescence, how does anti-EP8 perform in Western? Does it similarly detect the putative PS full-length and CTF bands? Can binding of anti-EP7 compete away with the blocking peptide? It would be most convincing to clone and overexpress T. cruzi PS heterologously e.g. in HEK cells to prove antibody specificity.
Previous author response: The decision to generate both anti-EP7 and anti-EP8 was to improve the chance of obtaining at least a single, high-quality anti-PS-like protein reagent. After our initial evaluations of their performance and specificity, we chose to focus entirely on anti-EP7 and did not test EP8 by western blot. While the expression of the T. cruzi PS-like protein would be an excellent confirmatory approach to evaluate specificity, the reality in Brazil is that it would be expensive, laborious, and take an undeterminable amount of time that would greatly delay publication. We believe the additional competition peptide experiment, presented above, demonstrates that the anti-serum is specific to the peptide for EP7 and when combined with the restricted presence of the sequence exclusively to the PS-like coding sequence provides sufficient confidence in the supposition that the antisera are only recognizing the PS-like protein.
New reviewer’s comment: While the competition data undoubtedly strengthen the good binding properties of anti-EP7 antiserum, these cannot rule out that this antiserum binds (an) other protein(s) in the T. cruzi whole cell extract. To rule this out, it would be critical to have a controlled setting with a proven higher/lower expression of the PS-like protein. Blotting of membrane fractions would also be helpful here. I have similar concerns regarding the immunostaining in Fig. 5.
New author response: A major benefit from our approach of identifying linear B-cell epitopes in our target protein prior to generating antibodies/antiserum is that we can evaluate the sequence through bioinformatics to ensure the highest possible specificity not only between species but also within the proteome of the target organism. As such, our analysis of EP7 and EP8 determined that they would exclusively represent the T. cruzi PS-like protein and no other. For this reason, our focus was on the performance of the antiserum and not its specificity. To convey this more clearly to the readers, we have included a sentence that reads (Pg 11, lines 373-374):
“Lastly, the uniqueness of each sequence to only represent the T. cruzi PS-like protein was evaluated to ensure specificity.”
